# Diffusion from convection

Marko Medenjak[1★], Jacopo De Nardis[2] and Takato Yoshimura[1,3]

**1** Institut de Physique Théorique Philippe Meyer, École Normale Supérieure,
PSL University, Sorbonne Universités, CNRS, 75005 Paris, France
**2** Department of Physics and Astronomy, University of Ghent,
Krijgslaan 281, 9000 Gent, Belgium.
**3** Department of Mathematics, King's College London, Strand, London WC2R 2LS, U.K.

★ medenjak@lpt.ens.fr

## Abstract

We introduce non-trivial contributions to diffusion constants in generic many-body systems with Hamiltonian dynamics arising from quadratic fluctuations of ballistically propagating, i.e. convective, modes. Our result is obtained by expanding the current operator in terms of powers of local and quasi-local conserved quantities. We show that only the second-order terms in this expansion carry a finite contribution to diffusive spreading. Our formalism implies that whenever there are at least two coupled modes with degenerate group velocities the system behaves super-diffusively, in accordance with non-linear fluctuating hydrodynamics. Finally, we show that our expression saturates the exact diffusion constants in quantum and classical interacting integrable systems, providing a general framework to derive these expressions.



# 1   Introduction

From its inception statistical physics has strived to derive the laws of hydrodynamics and thermodynamics. Its strength lies in the universality of the results, where only few aspects of microscopic system can influence the physics on macroscopic level. One of the outstanding open problems in the field is to explain the transport behavior of strongly interacting many-body systems by identifying the relevant degrees of freedom regulating the dynamics on hydrodynamical scales. While the emergence of ideal transport in interacting systems has been connected to the presence of *local conservation laws*, which prevent the decay of the current [1–3], much less is known about the microscopic origins of diffusive transport in Hamiltonian systems. Proving the emergence of diffusive transport and in particular computing diffusion constants in generic interacting many-body systems directly from their *Hamiltonian reversible dynamics* is still largely an open question [4]. This is a very non-trivial task even with the powerful numerical methods available for one-dimensional systems [5–7]. Recently, many non-trivial analytical results were obtained in holographic matter [8–12], random or noisy models [13–16] and calculated numerically in some generic chaotic systems [6]. The first analytical results on diffusion in reversible many-body classical systems date back to the second half of previous century and the studies of hard rod gasses [17–19]. The research has been reinvigorated in past years by the advances in the theory of quantum integrable systems [20–25], following the surprisingly discovery that in the presence of interactions the diffusive spreading occurs despite integrability.

     Regardless of these developments, a clear account of the mechanism leading to diffusion in integrable systems, that could provide the connection with the transport properties of generic systems, is still missing. In this article we fill this void by deriving a closed-form expression for the contribution to diffusion coefficients arising from the interaction of convective modes. While we require that the system exhibits convection, i.e. ballistic, transport for some degrees of freedom we make no assumptions about the integrability structure, which makes our theory applicable to general many-body systems with at least one conserved quantity [26], for instance quantum fluids with translational invariance symmetry [27, 28], non-integrable anharmonic chains [29], and non-integrable cellular automata [21, 30]. Surprisingly, the convective contribution to diffusion relies only on *stationary* properties of conservation laws and their currents.

The diffusion constant is obtained by the power series expansion of the current operator within the hydrodynamical cell in terms of conservation laws. A part of diffusion constant can then be related to the projection of the current onto the second-order term in this expansion. This contribution arises as a consequence of the dispersion of convective modes in thermal or non-thermal ensembles. On the level of the second-order term in this expansion, the Euler scale equations for the eigenmodes of the system can be used to derive a closed-form expression which saturates the diffusion constants in integrable theories [22, 23, 31]. A natural connection can also be made with a lower bound on the diffusion constant in terms of quadratically extensive quantities [32], and in terms of the curvature of Drude weights [33]. It should be stressed that we do not assume the presence of randomness in the dynamics

The approach is reminiscent of the non-linear fluctuating hydrodynamics (NLFHD) theory [28, 29, 34, 35], which employs the expansion of the currents in terms of conservation laws. Importantly, however, NLFHD is a phenomenological theory which cannot, at the moment, be used to evaluate the diffusion constant. The main accomplishment of NLFHD was to show that the presence of a quadratic coupling of the modes in the second-order expansion gives rise to Kardar-Parisi-Zhang [36] or Lévy super-diffusive universal transport. Generalization of this result manifests itself within the framework of our theory as a divergence of the diffusion constant in the presence of degenerate group velocities.

## 2 Diffusion in Hamiltonian many-body systems

Let us consider the Hamiltonian dynamics on an infinite chain $k \in \mathbb{Z}$. The conservation of energy implies continuity equation for energy density $h_k$,

$$\partial_t h_k(t) - j_{k+1}(t) + j_k(t) = 0. \tag{1}$$

For $t = 0$ or $k = 0$ the corresponding space/time coordinate will be omitted. Diffusion constant describes how a localized energy packet spreads through the system in thermal equilibrium, and can be therefore defined in terms of the variance of dynamical structure factor [31]

$$\frac{1}{C} \sum_y y^2 \langle h_y(t), h \rangle = \frac{D}{C} t^2 + \mathfrak{D} t + \mathcal{O}(1), \tag{2}$$

where $C$ is susceptibility, $D$ Drude weight, and $\mathfrak{D}$ the diffusion constant. Here we introduced a connected correlation function

$$\langle a, b \rangle = \langle ab \rangle - \langle a \rangle \langle b \rangle \tag{3}$$

with respect to the thermal average $\langle \bullet \rangle = \frac{\mathrm{tr}(\bullet \rho(\beta))}{\mathrm{tr}(\rho(\beta))}, \quad \rho(\beta) = \exp(-\beta H)$. Using the continuity equation and $\mathcal{PT}$ invariance of Hamiltonian dynamics one can relate the diffusion constant to the Onsager matrix $\mathfrak{L} = \mathfrak{D}C$, where the latter corresponds to the current-current correlation function [31]

$$\mathfrak{L} = \sum_x \int \mathrm{d}t (\langle j_x(t), j \rangle - D). \tag{4}$$

Importantly, the Onsager matrix is connected to the response of the current to the linear gradient of the external field in the Kubo formalism [37].

The above discussion can be trivially generalized to the systems with multiple conservation laws $\{q\}$ by introducing Drude weight, susceptibility, diffusion constant and Onsager matrices.

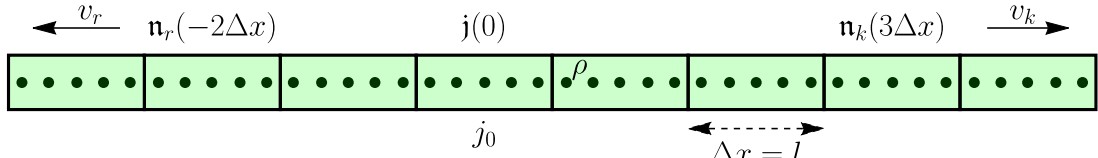

Figure 1: Schematic representation of the contribution to diffusion from convective modes. Current operator at origin excite two normal modes on top of the density matrix $\rho$, which traverse the system with fixed velocities. The normal modes give the contribution to the Onsager matrix when they reach the current operator at position $\chi$.

## 3  Hydrodynamics on the super-lattice

The goal of hydrodynamics is to identify relevant degrees of freedom which survive on large space-times scales, and thus naturally effect the diffusion constant and Drude weight (2). In order to identify hydrodynamic degrees of freedom the lattice is decomposed into the fluid cells of length $\Delta x = \ell$, and hydrodynamic densities are obtained as sums of (quasi)local operators $o_k$ within the fluid cell, and represented by the fraktur font $\mathfrak{o}(\chi, t) = \sum_{k=x-\ell/2+1}^{x+\ell/2} o_k(t)$, where $\chi = x/\ell$ is the rescaled coordinate. The operators which extend throughout the chain are denoted by capital letters $O(t) = \sum_\chi \mathfrak{o}(\chi, t)$

The main *hydrodynamic assumption* asserts that in the large time limit many-body systems locally equilibrate [38], and that the complete information about the dynamics is contained in local conserved quantities $Q_i = \sum_\chi \mathfrak{q}_i(\chi)$, with $i = 1, \ldots, n_c$. This leads us to introduce the set of locally equilibrated maximum entropy ensembles

$$\rho(\underline{\beta}(\chi)) = \exp(\beta^i(\chi)\mathfrak{q}_i(\chi)), \tag{5}$$

with temperatures $\underline{\beta} = (\beta^1, \beta^2, \cdots)$ and Einstein's repeated indices summation convention $a^i(\chi)\mathfrak{o}_i(\chi) = \sum_{i=1}^N \sum_\chi a^i(\chi)\mathfrak{o}_i(\chi)$. The average, $\langle \bullet \rangle = \frac{\text{tr}(\bullet \rho(\underline{\beta}(\chi)))}{\text{tr}(\rho(\underline{\beta}(\chi)))}$, is always taken with respect to some homogeneous density matrix $\beta^k(\chi) = \beta^k$, except if expression involves derivatives with respect to temperatures $\beta^k(\chi)$, or expectation value of the charge $\mathfrak{q}_i(\chi) = \langle \mathfrak{q}_i(\chi) \rangle$, in which case, the homogeneous limit is taken after derivatives.

Since the transport coefficients are related to the asymptotic behavior of two point functions we will consider the following scaling limit $x = \chi \times \ell$ and $t = \tau \times \ell$ with $\ell \to \infty$. The hydrodynamical assumption asserts that in the scaling limit any operator $\mathfrak{o}(\chi)$ can be expressed as a function of conserved charges $\mathfrak{o}(\chi) = f(\underline{\mathfrak{q}})$ and their powers, which is related to local equilibration. There have recently been many works showing that in the homogeneous setup equilibration to the state depending only on conservation laws occurs [39]. Up to the second order the expansion in terms of local charges takes the form

$$\delta\mathfrak{o}(\chi) = (\partial_{\mathfrak{q}_i(\chi)}\langle\delta\mathfrak{o}(\chi)\rangle)\delta\mathfrak{q}_i(\chi) + \frac{1}{2}(\partial_{\mathfrak{q}_j(\chi)}\partial_{\mathfrak{q}_i(\chi)}\langle\delta\mathfrak{o}(\chi)\rangle)\delta\mathfrak{q}_i(\chi)\delta\mathfrak{q}_j(\chi) + \mathcal{R}, \tag{6}$$

with $\delta\mathfrak{o} = \mathfrak{o} - \langle\mathfrak{o}\rangle$, where $\delta\mathfrak{q}_{i_k}$ correspond to the densities of (quasi)local conservation laws. The expansion (6) admits an immediate physical interpretation: it corresponds to the variation of the stationary expectation value of observable with respect to the expectation values of conserved quantities. Simply put, on the hydrodynamical scale the perturbation of local equilibrium by an operator $\mathfrak{o}$ can be obtained by projecting the operator on corresponding charges [40]. In particular, we can verify that such an expansion satisfies a consistency condition on the level of two point functions $\langle\delta\mathfrak{o}_1, \delta\mathfrak{o}_2\rangle$ obtained by expanding only $\delta\mathfrak{o}_1$ or both

operators $\delta\mathfrak{o}_1$ and $\delta\mathfrak{o}_2$ S1.2. The remainder terms $\mathcal{R}$ can include non-local charges, higher order contributions, and contributions from charges within neighbouring cells as outlined in S4. In what follows we will focus solely on the contribution to diffusion constant arising from the second order of our expansion, i.e. the convective modes.

In order to determine the contribution to diffusion constant from expansion (6), we have to deduce the dynamics of the second order term in the leading order in $\ell$, i.e. *Euler scale*. This is easily obtained by solving the continuity equation $\partial_\tau \mathfrak{q}(\chi,t) + (\mathfrak{j}(\chi+1,\tau) - \mathfrak{j}(\chi,\tau)) = 0$, using only the first order contribution from the expansion of the current $\mathfrak{j}$. The solution of the linear differential equations can be expressed in terms of *normal modes* $\mathfrak{n}_i$, which are orthonormal linear combinations of charges $\mathfrak{q}_i = (R^{-1})_i^{\ j} \mathfrak{n}_j$ [35]. Orthonormality condition implies that $(RCR^\mathrm{T})_{ij} = \delta_{ij}$, where $C_{ij} = \frac{1}{\ell}\langle \mathfrak{q}_i(0), \mathfrak{q}_j(0)\rangle$ is the susceptibility matrix. Physically, normal modes correspond to the localized wave-packets on top of the density matrix $\rho(\beta)$ which move with distinct velocities $v_k$. This means that in the Fourier space $\hat{n}_i(k,\tau) = \sum_\chi e^{-\overline{\mathrm{i}k}\chi} n(\chi,\tau)$ they satisfy the continuity equation

$$\partial_\tau \hat{n}_i(k,\tau) + \mathrm{i}\omega_i(k)\hat{n}_i(k,\tau) = 0, \tag{7}$$

with $\omega_i(k) = v_i k$. Equation (7) admits the corrections of the order $\mathcal{O}(\ell^{-1})$, due to the spreading and diffusion of wave-packets, which we here disregard as we are interested in the convective contribution to diffusion constant (see also [40]).

## 4  Drude weights

To demonstrate the utility of our expansion (6), we employ it to obtain the *Drude weights*, namely the coefficients parametrising the ballistic direct conductivities [41, 42], which are defined as $D_{kl} = \lim_{t\to\infty}(2t)^{-1}\int_{-t}^{t}\mathrm{d}s \sum_x \langle j_{k,0}(s), j_{l,x}(0)\rangle$, for some mode $k$ and $l$. In terms of the hydrodynamical currents the matrix of Drude weights reads

$$D_{kl} = \lim_{\tau\to\infty}\frac{1}{2\tau\ell}\sum_\chi \int_{-\tau}^{\tau}\mathrm{d}\tau' \langle \mathfrak{j}_k(0,\tau'), \mathfrak{j}_l(\chi,0)\rangle. \tag{8}$$

The only contribution to the Drude weight (8) that remains finite after the hydrodynamical expansion (6) in the limit $\ell \to \infty$ corresponds to the linear order, which reproduces a well-known result [19, 43] $D_{kl} = (\partial_{\mathfrak{q}_i(0)}\langle \mathfrak{j}_k(0)\rangle)\langle Q_i, j_{l,0}\rangle = BC^{-1}B$, with $B_{ik} = \langle \mathfrak{q}_i(0), \mathfrak{j}_k(0)\rangle_n$. In the normal mode basis the Drude weights read

$$D_{kl} = \langle j_k, N^k\rangle\langle N_k, j_l\rangle, \tag{9}$$

where we introduced extensive normal modes $N_i = R_i^{\ j}Q_j$. The physical interpretation of this result is the following, see also Fig. 1. The current operator at the origin $\mathfrak{j}_l(0)$ creates excitations on top of the density matrix $\rho(\beta)$. The only stable excitations, i.e. excitations which survive the hydrodynamical limit $\ell \to \infty$, and which are able to reach the current operator at point $\chi$, $\mathfrak{j}_l(\chi)$, are the densities of conserved charges. In the normal mode basis excitation has a well defined velocity $v_k$ and gives a contribution to the Drude weight when it reaches the current operator $\mathfrak{j}_l(\chi)$.

## 5  Diffusion constants

As already explained the Onsager coefficients are related to the diffusion constant via Einstein's relation $\mathfrak{L}_{kl} = \mathfrak{D}_k^{\ j}C_{jl}$ [31, 44]. We can represent it compactly in terms of the sub-ballistic

current $j_k^-(\chi,\tau) = j_k(\chi,\tau) - (\partial_{q_i(\chi)}\langle j_k(\chi)\rangle)q_i(\chi,\tau)$, as

$$\mathfrak{L}_{kl} = \sum_\chi \int d\tau \langle j_k^-(\chi,\tau), j_l^-(0,0)\rangle. \tag{10}$$

Plugging the expansion for the current (6), and the Euler scale dynamics of charges (7) produces a finite *convective* contribution $\mathfrak{L}_{kl}^c$ to the Onsager matrix, following a lengthy but elementary manipulation (see S2.2 for details)

$$\mathfrak{L}_{kl}^c = 2(R^{-1}\tilde{G}^2 R^{-T})_{kl}, \tag{11}$$

where we introduced the renormalized coupling coefficient

$$\tilde{G}_{ij}^2 = \frac{G_{ii'j'}G_j^{i'j'}}{|v_{i'} - v_{j'}|}, \tag{12}$$

expressed in terms of the quadratic matrix $G_i^{jk} = R_i^l((R^{-1})^T H_l R^{-1})^{jk}/2$ and Hessian $H_v^{ij} = \ell\partial_{q_i(0,0)}\partial_{q_j(0,0)}\langle j_v(0,0)\rangle$.

Once again the result admits a simple physical interpretations (reminiscent of a kinetic-like picture [23]). At time $\tau$ the nonzero contribution is produced by excitations created by the currents $j(\chi,\tau)$, that reach the operator at the origin $j(0,0)$. The weight of contribution is given by the overlap of two normal modes and the current, see FIG 1. To be more quantitative the Hessian $H_v^{ij}$, is rotated by $R$'s when changing the basis from charge densities $q_i$ to normal modes $n_i$, while the renormalization $|v_{i'} - v_{j'}|$ arises from the scattering of two normal modes $n_{i'}$ and $n_{j'}$ with current: $\int d\tau e^{-i(\omega_{i'}(k) + \omega_{j'}(-k))\tau} = \frac{2\pi\delta(k)}{|v_{i'} - v_{j'}|}$. The convective Onsager matrix can again be nicely represented in a closed form in terms of normal modes (see S2.2 and also [40])

$$\mathfrak{L}_{kl}^c = \frac{\langle j_k^-, N_i N_j\rangle\langle N^i N^j, j_l^-\rangle}{2|v_i - v_j|}. \tag{13}$$

In what follows we will check the validity and new implications of our result (13), by comparing its predictions with previous results.

## 6    Lower bounds on diffusion

The diagonal elements of Onsager matrix $\mathfrak{L}_{kk}^c$ are generally expected to be a lower bound for the exact ones. There have been several proposals in the past years to lower bound the diffusion coefficients. Here we establish a direct connection between our expression (11) and two recent results.

The first bound on diagonal diffusion coefficients $\mathfrak{L}_{k,k}^s \geq (\langle J_k, Q\rangle)^2/(8v_{LR}\langle Q, Q\rangle)$ at infinite temperature $\langle \bullet \rangle = \frac{tr(\bullet)}{tr(\mathbb{1})}$ was derived in [32]. Here $v_{LR}$ is the Lieb-Robinson velocity and $Q$ is a conserved quantity which scales quadratically $\langle Q, Q\rangle \sim L^2$ with the system size $L$. A set of such conservation laws can be obtained by multiplying two local integrals of motion $Q = \sum_{i\geq j}\alpha_{ij}N_i N_j$, where we choose traceless $N_i$. Provided that all of the quadratically extensive quantities are of this form, we can show that our expression supersedes this lower bound for arbitrary values of coefficients $\alpha_{ij}$ (see S3 for the details). However, additional quadratically extensive quantities, not given by product of local and quasi-local charges, can in principle exist. These terms, included in the extra contribution $\mathcal{R}$ in (6), would give an extra contribution to the full Onsager coefficients $\mathfrak{L}_{kk}$.

The second lower bound on the spin/charge diffusion constant in the zero magnetization sector/half-filling has been proposed in [33] and further studied in [25, 45]. It corresponds to the curvature of the Drude weight with respect to the magnetization $\langle S^z \rangle$ filling $v(T, h) = 4T \langle S^z \rangle$ as $\mathfrak{L}_{s,s} \geq \partial_v^2 D(h)/v_{\mathrm{LR}}$. As a consequence of the spin flip symmetry in the considered examples, one of the normal modes in our expression for the diffusion constant (13) should be magnetization $N_i = S^z$. Since the velocity of magnetization normal mode vanishes, the lower bound can be reproduced by replacing the velocity in our expression (13) with their upper bound $v_{\mathrm{LR}}$.

# 7  Diffusion in integrable systems

Following recent developments in the hydrodynamic description of integrable systems [23, 31, 46, 46–51], the transformation $R$ as well as other transport matrices such as $A, B$, and $G$ can be computed exactly. In integrable systems normal modes are stable quasi-particle excitations that fully describe the thermodynamics and hydrodynamics of the system [52, 53]. An expression for the $\mathfrak{L}_{ij}$ matrix has been found in [31] by exploiting integrability techniques. It is not hard to see that our result reproduces this expression exactly $\mathfrak{L}_{ij}^c \equiv \mathfrak{L}_{ij}$ S2.3. Notice that the same applies for the diffusion matrices of hard rod gases [19]. We can therefore conclude that diffusive transport in integrable models and hard rods gases is given *purely* by the dispersion of their ballistic modes.

Our result also gives a simple explanation of the curious "magic formula", which was established rigorously at infinite temperatures [54], and numerically at finite temperatures. The "magic formula" relates the curvature of self-Drude weight $D_{s,s}^{\mathrm{self}} = \int \mathrm{d}t \langle j_s(0, t) j_s(0, 0) \rangle^c$ where $j_s$ is the spin (or charge) current, with the spin diffusion constant $\mathfrak{D}_{s,s} = \mathfrak{L}_{s,s}/C_{s,s} = \partial_v^2 D_{s,s}^{\mathrm{self}}(v)$. This is now readily understood by noticing that the curvature of the self-Drude weight is equivalent to equation (11) with $\mathfrak{q}_j = S^z$ and $j_k = j_s$, and that in integrable models $\mathfrak{L}_{kl}^c \equiv \mathfrak{L}_{kl}$.

# 8  NLFHD and super-diffusion:

NLFHD is based on expanding the expectation values of the *extensive* currents in terms of the local conservation laws up to the second-order and adding phenomenological diffusion constants and white noise terms related by the fluctuation-dissipation relation [28, 29, 34, 35]. Since the phenomenological diffusion constant and white noise affects diffusive processes, NLFHD has no predictive power on diffusive scales. However, in the presence of a single conserved charge one obtains a noisy Burgers' equation, provided that the current has an overlap with the square of the charge. Burgers' equation is known to belong to the KPZ universality class with dynamical exponent $z = \frac{3}{2}$ [36, 55]. This implies that diffusion constant diverges $\mathfrak{L} = \infty$. Similar behavior can be identified in the presence of multiple modes, and detailed analysis reveals a plethora of new super-diffusive universality classes arising in the presence of appropriate multi-mode couplings [35]. This behavior can be understood within our framework, since the self-coupling term corresponds to the presence of non-zero matrix elements with degenerate velocities (12), and results in the divergence of the convective Onsager coefficients (11). More specifically, the diffusion constant of a mode diverges whenever the mode coupling matrix $G_i^{\ jk}$ has non-zero elements for at least two modes with degenerate velocities. Such situations were typically exempted from past NLFHD applications, where the non-degeneracy of velocities was usually assumed in order to justify the mode decoupling assumption. Such situations are, however, expected to be important in integrable chains, where super-diffusion can also be observed for the spin or charge degrees of freedom close to half-filling [56–59].

# 9    Conclusion

We have introduced an operatorial expansion of the currents in a many-body system in terms of hydrodynamical densities of conservation laws in generic stationary states. We have shown that the second-order terms of this expansion give rise to a finite contribution to the Onsager matrix and therefore to the diffusion constants of the system. Our framework unifies previous results on diffusive transport in one dimension and shows that in integrable systems convective contributions saturate the exact diffusion constants, demonstrating that other mechanisms are absent. Nevertheless, convective contributions account only for a part of the full diffusion constant in classical probabilistic dynamical systems [21,30], and are completely absent in the spin chain with strong dephasing [60]. These partial results call for clarification of which contributions are non-vanishing in certain systems and whether generic Hamiltonian systems support non-trivial quadratic charges that are not included in the convective part of diffusion.

For the case of integrable models we provided another non-trivial verification of the expression for the diffusion constants and proved the so-called magic formula. Moreover, we have shown how the diffusion constant of a certain charge can diverge whenever there is degeneracy of the velocity of the modes. While this mechanism is valid for generic systems, it might prove to be useful to explain the emerging KPZ super-diffusive dynamics of spin and charge in quantum and classical isotropic chains [54,61,62]. In these models the spin/charge mode at half-filling has zero velocity and it constitutes an accumulation point for all other modes with finite velocities [63] that approach zero as $v_i \equiv v_{\theta,a} \sim a^{-1}$ [45,61] for any $\theta$. An interesting direction for future research is to develop these ideas on a more rigorous footing.

While we here explored the effects of convective modes on the Onsager matrix, they can also contribute on the level of higher-order cumulants of the currents. Our construction indicates that the contributions are hierarchically ordered with respect to the order of the cumulant, analogously to the BBGKY hierarchy in the Boltzmann equation [64,65]. Finally, our expansion can also be applied in higher dimensional systems, where normal diffusion is typically found. It would be interesting to compare our convective contribution with the known results [66].

## Acknowledgements

We thank B. Doyon for numerous fruitful exchanges, and would like to refer to his complementary publication [40]. We would also like to thank D. Bernard, B. Doyon, E. Ilievski for useful comments on the manuscript. J.D.N. is supported by Research Foundation Flanders (FWO). TY acknowledges the support by Takenaka Scholarship Foundation, and hospitality at Tokyo Institute of Technology.

# S1    Extra details on derivation of hydrodynamics from the lattice

## S1.1    Correlations on super-lattice

In order to identify the effects of multi-point correlation functions on transport coefficients we consider an infinitely large spin chain, which we divide into the parts of equal length $\Delta x = \ell$, i.e. the hydrodynamical cells. We will denote the macroscopic coordinate which determines the position on this supper-lattice by $\chi$, $\chi = x \times \ell$.

First of all, we will show that multi-point correlation functions of hydrodynamic density of the local operator and conservation laws scale linearly with the size of the hydrodynamic cell,

if all of the operators are located within the same hydrodynamic cell, and are at most constant otherwise. This result is necessary to establish that the contributions from the beyond nearest neighboring cells, and higher orders in the expansion cannot contribute to the Onsager matrix.

The first part of the result follows directly from the clustering property of GGE's and the fact that the cells that are not nearest neighbors get separated with increasing $\ell$. The second part of the result, follows from simply noticing that the higher point connected correlators can be interpreted as a derivative of expectation values, which scales linearly with the size of the cell.

The hydrodynamical density $\mathfrak{o}(\chi)$ is given by

$$\mathfrak{o}(\chi) = \sum_{i=\chi-\ell/2+1}^{\chi+\ell/2} o(i), \tag{S1}$$

in terms of the quasilocal operator

$$o(i) = \sum_{k=0}^{\infty} o_{[i,i+k]}. \tag{S2}$$

Quasilocality means that the spectral norm is upper bounded by

$$\|o_{[i,i+k]}\| < \alpha_1 \times \exp(-\eta_1 k), \tag{S3}$$

for some $\alpha, \zeta > 0$, and the $C^*$ norm $\|\bullet\|$. The operators $o_{[i,i+k]}$, are supported on the sublattice $[i, i+k]$, and act as an identity operator everywhere else.

The multi-point connected correlation function is defined as

$$\langle \mathfrak{o}(\chi) \mathfrak{q}^1(\chi_1) \cdots \mathfrak{q}^N(\chi_N) \rangle^c \equiv \frac{\partial^N \langle \mathfrak{o}(\chi) \rangle}{\partial \beta^1(\chi_1) \cdots \partial \beta^N(\chi_N)}, \tag{S4}$$

with the expectation value $\langle \bullet \rangle = \frac{\mathrm{tr}(\bullet \rho(\underline{\beta}(\chi)))}{\mathrm{tr}(\rho(\underline{\beta}(\chi)))}$ in a generalized, locally thermalized state $\rho(\underline{\beta}(\chi)) = \exp(\beta^i(\chi)\mathfrak{q}_i(\chi))$. Assuming that all of conserved densities commute up to the boundary terms, the two point connected correlation function $\langle \mathfrak{a}, \mathfrak{q} \rangle = \langle \mathfrak{a}\mathfrak{q} \rangle^c$. We are considering a set of states $\rho$ which satisfy exponential clustering

$$\langle a_{[i,j]} b_{[k,l]} \rangle^c < \alpha_2 \exp(-\eta_2 |j-k|) \|a_{[i,j]}\| \times \|b_{[k,l]}\|, \tag{S5}$$

where we assumed the ordering $i \le j \le k \le l$. It is not hard to see that the pairs of hydrodynamical operators decay exponentially with the distance

$$|\langle \mathfrak{o}_1(\chi), \mathfrak{q}_2(\chi') \rangle| \le \alpha_3 \exp(-\eta_3 \ell), \tag{S6}$$

if $|\chi - \chi'| > 2$, i.e. the operators are not located in the same, or the neighboring cells. In order to demonstrate the property (S6) we can divide the connected correlator into two contributions

$$|\langle \mathfrak{o}_1(\chi), \mathfrak{q}_2(\chi') \rangle| \le \sum_{i=x-\ell/2+1}^{x+\ell/2} \sum_{k=0}^{\ell/2} |\langle \mathfrak{o}_{1[i,i+k]}, \mathfrak{q}_2(\chi') \rangle| +$$
$$+ \sum_{i=x-\ell/2+1}^{x+\ell/2} \sum_{k=\ell/2+1}^{\infty} |\langle \mathfrak{o}_{1[i,i+k]}, \mathfrak{q}_2(\chi') \rangle| = s_1 + s_2.$$

In order to upper bound the second term $s_2$ we can use the trivial bound

$$\langle a, b \rangle \le \|a\| \times \|b\|, \tag{S7}$$

and quasilocality (S3)

$$s_2 \le \mathrm{Const.} \times \ell^2 \exp(-\eta_1 \ell/2). \tag{S8}$$

In order to evaluate the first term, we take into account that $|\chi - \chi'| > 2$, implying that the minimal distance between the operators in $\mathfrak{o}_1$ and densities $\mathfrak{q}_2$ is $\frac{\ell}{2}$. This enables us to show that

$$s_1 \leq \text{Const.} \times \ell^3 \exp(-\eta_2 \ell/2). \tag{S9}$$

Now we proceed to show that the contribution from the neighboring cell is sub-polynomial in the size of hydrodynamic cell $\ell$

$$|\langle \mathfrak{o}_1(\chi), \mathfrak{q}_2(\chi+1)\rangle| \leq \text{Const.} \times \ell^{\kappa}, \tag{S10}$$

for arbitrary $\kappa$. We divide the sum into five parts

$$|\langle \mathfrak{o}_1(\chi), \mathfrak{q}_2(\chi+1)\rangle| \leq$$
$$\leq \sum_{i=x-\ell/2+1}^{x+\ell/2-\ell^{\kappa_1}} \left( \sum_{k=0}^{\ell^{\kappa_2}} |\langle o_{1[i,i+k]}, \mathfrak{q}_2(\chi+1)\rangle| + \sum_{k=\ell^{\kappa_2}}^{\infty} |\langle o_{1[i,i+k]}, \mathfrak{q}_2(\chi+1)\rangle| \right) +$$
$$+ \sum_{i=x+\ell/2-\ell^{\kappa_1}+1}^{x+\ell/2} \sum_{k=0}^{\ell^{\kappa_2}} \left( \sum_{j=x+\ell/2+1}^{x+\ell/2+\ell^{\kappa_1}} |\langle o_{1[i,i+k]}, q_{2,j}\rangle| + \sum_{j=x+\ell/2+\ell^{\kappa_1}}^{x+3\ell/2} |\langle o_{1[i,i+k]}, q_{2,j}\rangle| \right) +$$
$$+ \sum_{i=x+\ell/2-\ell^{\kappa_1}+1}^{x+\ell/2} \sum_{k=\ell^{\kappa_2}}^{\infty} |\langle o_{1[i,i+k]}, \mathfrak{q}_2(\chi+1)\rangle| = s_1 + s_2 + s_3 + s_4 + s_5,$$

with $0 < \kappa_2 < \kappa_1$. Similarly as before we can lower bound the second sum by using a trivial lower bound (S7) and quasilocality

$$s_2, s_5 \leq \text{Const.} \times \ell^2 \exp(-\eta_1 \ell^{\kappa_2}). \tag{S11}$$

Using similar arguments as before, we get

$$s_1, s_4 \leq \text{Const.} \times \ell^{\kappa_2+2} \exp(-\eta_2(\ell^{\kappa_1} - \ell^{\kappa_2})). \tag{S12}$$

Using a trivial bound (S7), we can upper bound $s_3$ by

$$s_3 \leq \text{Const.} \times \ell^{\kappa_2+2\kappa_1}. \tag{S13}$$

Since $\kappa = 2\kappa_1 + \kappa_2$ can be arbitrarily small, we arrive at the result (S10).

We are now in the position to prove extensivity and orthogonality of arbitrary multipoint connected correlation function on the super-lattice

$$\langle \mathfrak{o}_1(\chi_1)\mathfrak{q}_2(\chi_2)\cdots\mathfrak{q}_N(\chi_N)\rangle^c = \ell(\langle \mathfrak{o}_1(\chi_1)\cdots\mathfrak{q}_N(\chi_1)\rangle_n^c \delta_{\chi_1\chi_2}\delta_{\chi_2\chi_3}\cdots\delta_{\chi_{N-1}\chi_N} + \mathcal{O}(\ell^{-1})). \tag{S14}$$

The $N$-point connected correlation function is at most extensive, since the expectation value $\langle \mathfrak{o} \rangle$ is proportional to the volume of the cell $\propto \ell$. And the $N$-point correlation function corresponds to the $N-1$ point derivative of the expectation value. In the absence of phase transitions divergences are absent, implying extensivity of $N$-point correlation function.

If the distance between at least two operators in the connected correlation function is $|\chi - \chi'| > 2$, the correlation function vanishes exponentially in the hydrodynamical limit $\ell \to \infty$. Let's assume that $\mathfrak{o}_i(\chi)$ and $\mathfrak{q}_j(\chi)$ do not occupy the same or the neighboring cells. Using (S6) we have that

$$|\langle \mathfrak{o}_i(\chi), \mathfrak{q}_j(\chi')\rangle| \leq \exp(-\eta_2(\beta_1(\chi), ..., \beta_N(\chi_N))\ell), \tag{S15}$$

and taking the derivatives produces at most polynomial factor $\ell^{N-2}$.

A technical result that we will need in next section is a hydrodynamical decomposition of the two point connected correlation function of squares of hydrodynamical densities into the products of two point functions of local hydrodynamical densities

$$\langle \delta\mathfrak{q}_1\delta\mathfrak{q}_2, \delta\mathfrak{q}_3\delta\mathfrak{q}_4 \rangle = \langle \delta\mathfrak{q}_1, \delta\mathfrak{q}_3 \rangle\langle \delta\mathfrak{q}_2, \delta\mathfrak{q}_4 \rangle + \langle \delta\mathfrak{q}_1, \delta\mathfrak{q}_4 \rangle\langle \delta\mathfrak{q}_2, \delta\mathfrak{q}_3 \rangle + \mathcal{O}(\ell), \tag{S16}$$

which we are going to prove only at infinite temperature but holds for any clustering density matrix $\rho$. The result can be inferred by explicitly decomposing the four point function into the sum of local terms

$$\langle \delta q_1 \delta q_2 \delta q_3 \delta q_4 \rangle\rangle_\infty = \sum_{\alpha_1,\alpha_2,\alpha_3,\alpha_4} \sum_{r_1,r_2,r_3,r_4} \frac{\text{tr}(\delta q_1^{[\alpha_1,\alpha_1+r_1]}\delta q_2^{[\alpha_2,\alpha_2+r_2]}\delta q_3^{[\alpha_3,\alpha_3+r_3]}\delta q_4^{[\alpha_4,\alpha_4+r_4]})}{\text{tr}(\mathbb{1})}.$$
(S17)

In order to evaluate the above sum we divide it into two parts. The first part corresponds to the case in which at least three of the densities overlap, and thus form a connected cluster. Such term will yield a finite contribution, only if all four densities form a connected cluster. If the largest support of the density in the above sum is $r_\beta = \max(r_1, r_2, r_3, r_4)$, then the absolute value of the sum over $\alpha_k$, $k \neq \beta$ of such terms can be upper bounded by

$$\sum_{k\neq\beta}\sum_{\alpha_k}\left|\frac{\text{tr}(\delta q_1^{[\alpha_1,\alpha_1+r_1]}\delta q_2^{[\alpha_2,\alpha_2+r_2]}\delta q_3^{[\alpha_3,\alpha_3+r_3]}\delta q_4^{[\alpha_4,\alpha_4+r_4]})}{\text{tr}(\mathbb{1})}\right| \leq$$
$$\leq r_\beta^3 \exp(-\eta(r_1+r_2+r_3+r_4)), \quad \eta > 0.$$

Furthermore we can upper bound $r_\beta < (r_1 + r_2 + r_3 + r_4)$, implying that the summing over $r_1, r_2, r_3, r_4$ yields a finite contribution. The only summation that remains is the one over $\beta$. This produces a factor which is proportional to the cell size $\ell$.

In order to consider remaining contributions, we have to take into account the cases, where none of the operators overlap at any point, and the case where exactly two operators overlap. In the first case the contribution automatically vanishes due to the tracelessness of $\delta q$. In order to compactly represent the second contribution we perform the expansion of the product of the two point correlation function

$$\langle \delta q_1, \delta q_3 \rangle \langle \delta q_2, \delta q_4 \rangle =$$
$$= \sum_{\alpha_1,\alpha_2,\alpha_3,\alpha_4} \sum_{r_1,r_2,r_3,r_4} \frac{\text{tr}(\delta q_1^{[\alpha_1,\alpha_1+r_1]}\delta q_2^{[\alpha_2,\alpha_2+r_2]})\,\text{tr}(\delta q_3^{[\alpha_3,\alpha_3+r_3]}\delta q_4^{[\alpha_4,\alpha_4+r_4]})}{\text{tr}(\mathbb{1})^2}.$$
(S18)

Using equivalent arguments as above, we can show that the terms which overlap scale linearly with the system size $\ell$.

This immediately implies that the trace of the product of four extensive operators can be represented as the sum of two point functions up to corrections of the order $\mathcal{O}(\ell)$, since the contributions from the operators corresponding to the overlapping of the support of two pairs of operators in the four point function produces exactly the contributions of the form (S18) up to the overlaping terms. This results in the decomposition (S16). Note that this expression is divided by $\ell^2$, and disregarded terms result in the $\frac{1}{\ell}$ correction to the diffusion constant.

## S1.2  Consistency condition

While in the main text a physical argument for the form of expansion was given, the coefficients can be obtained from the consistency condition.

The starting point is the local expansion in which we assume that the higher order terms in expansion

$$\delta\mathfrak{o} = c^i \delta\mathfrak{q}_i + \frac{c^{ij}}{\ell}\delta(\mathfrak{q}_i\mathfrak{q}_j) + \frac{c^{ijk}}{\ell^2}\delta(\mathfrak{q}_i\mathfrak{q}_j\mathfrak{q}_k) + ...,$$
(S19)

take the form that reproduces the connected correlation function, i.e. $\langle \mathfrak{o}, \delta(\mathfrak{q}_{i_1}\cdots\mathfrak{q}_{i_k})\rangle \equiv \langle \mathfrak{o}_1\mathfrak{q}_{i_1}\cdots\mathfrak{q}_{i_k}\rangle^c \leq \text{Const.} \times \ell$. In order for the expansion (S19) to be consistent, we have to require that the expansion $\delta\mathfrak{o}_k(\chi) = (c^i_k\delta\mathfrak{q}_i(\chi) + \frac{c^{ij}_k}{\ell}\delta\mathfrak{q}_i(\chi)\delta\mathfrak{q}_j(\chi) + ...)$

is consistent on the level of two point functions. In particular the leading order of the two point correlation function scales linearly with $\ell$ and reads

$$c_1^i \langle \mathfrak{o}_2, \delta \mathfrak{q}_{j_2} \rangle = c_1^i c_2^j \langle \delta \mathfrak{q}_i, \delta \mathfrak{q}_j \rangle. \tag{S20}$$

Note also that $\delta(\mathfrak{q}_i \mathfrak{q}_j) = \delta \mathfrak{q}_i \delta \mathfrak{q}_j$. Demanding that the above equality is satisfied for all operators $\mathfrak{o}_1$ (setting $c^{i_1'} = \delta_{i_1, k}$ in particular) the coefficients can be obtained easily by inverting the relation (S20)

$$c_1^i = \lim_{\ell \to \infty} \frac{1}{\ell} C^{ij} \langle \mathfrak{o}, \delta \mathfrak{q}_j \rangle. \tag{S21}$$

After solving the leading order equations, this contribution should be subtracted in expansion (S19) in order to eliminate the terms corresponding to the first order. The equations for the second order can then be obtained by taking the limit $\ell \to \infty$, which removes the higher order terms

$$\frac{c_1^{ij}}{\ell} \langle \mathfrak{o}^-, \delta \mathfrak{q}_i \delta \mathfrak{q}_j \rangle = \frac{c_1^{ij} c_2^{kl}}{\ell^2} \langle \delta \mathfrak{q}_i \delta \mathfrak{q}_j, \delta \mathfrak{q}_k \delta \mathfrak{q}_l \rangle. \tag{S22}$$

Taking into account the decomposition of the four point function $\langle \delta \mathfrak{q}_i \delta \mathfrak{q}_j, \delta \mathfrak{q}_k \delta \mathfrak{q}_l \rangle$ into two point functions (S16) and symmetry property of coefficients $c_r^{ij} = c_r^{ji}$ for $r \in \{1, 2\}$, one readily recovers the result

$$c_2^{ij} = \tfrac{1}{2} \lim_{\ell \to \infty} \frac{1}{\ell} C^{ik} C^{jl} \langle \mathfrak{o}^-, \delta \mathfrak{q}_k \delta \mathfrak{q}_l \rangle, \tag{S23}$$

by following the first order prescription.

## S2 Extra Derivations of equations

### S2.1 Derivation of Drude weights

As noted in the main text in order to derive the Drude weight and diffusion constant, we will use the dynamics of normal modes on Euler scale

$$\mathfrak{n}_i(\chi, \tau) = \sum_{\chi'} \frac{1}{2\pi} \int_{-\pi}^{\pi} dk\, e^{ik(\chi - \chi') - i\omega_i(k)\tau} \mathfrak{n}_i(\chi') + \ldots, \tag{S24}$$

up to $\frac{1}{\ell}$ corrections, and the hydrodynamic expansion

$$\delta \mathfrak{o}(\chi) = (\partial_{\mathfrak{q}_i(\chi)} \langle \delta \mathfrak{o}(\chi) \rangle) \delta \mathfrak{q}_i(\chi) +$$
$$+ \frac{1}{2} (\partial_{\mathfrak{q}_j(\chi)} \partial_{\mathfrak{q}_i(\chi)} \langle \delta \mathfrak{o}(\chi) \rangle) \delta \mathfrak{q}_i(\chi) \delta \mathfrak{q}_j(\chi) + \mathcal{R}, \tag{S25}$$

for the current.

In order to establish how the current expansion works in actual computations, we will apply it to the computation of Drude weight. The Drude weights $D_{i,j}$ are defined by

$$D_{i,j} = \lim_{\tau \to \infty} \frac{1}{2\tau \ell} \sum_{\chi} \int_{-\tau}^{\tau} d\tau' \langle \mathfrak{j}_i(0, \tau'), \mathfrak{j}_j(\chi, 0) \rangle. \tag{S26}$$

For computing this object, we need the first order in the hydrodynamic expansion (S25) only.

Inserting this term into (S26) gives

$$
\begin{aligned}
D_{i,j} &= \lim_{\tau\to\infty} \frac{1}{2\tau\ell} \sum_{\chi} \int_{-\tau}^{\tau} d\tau' \frac{\partial \langle \mathsf{j}_i(0,0)\rangle}{\partial \mathsf{q}_k(0,0)} \langle \mathsf{q}_k(0,\tau'), \mathsf{j}_j(\chi,0)\rangle = \\
&= \lim_{\tau\to\infty} \frac{1}{2\tau\ell} \sum_{\chi} \int_{-\tau}^{\tau} d\tau' \frac{1}{\ell} C^{kl} \langle \mathsf{j}_i(0,0), \mathsf{q}_k(0,0)\rangle \langle \mathsf{q}_l(0,\tau'), \mathsf{j}_j(\chi,0)\rangle = \\
&= (BC^{-1}B)_{i,j},
\end{aligned}
\tag{S27}
$$

where we used the relation $\partial_{\beta^i(x)} = \ell C_{ij}\partial_{\mathsf{q}_j(x)}$, which follows from the clustering property (S14). To go from the second to the third line, one should notice that due to the homogeneity of the stationary state, the space dependence of the current $\chi$ can be moved to the charge $\mathsf{q}_l(0,\tau')$. Summing over the spatial coordinate $\chi$ results in a conserved quantity, allowing us to drop the time dependence $\tau'$.

Alternatively, changing to the normal mode basis using the convention $RCR^{\mathrm{T}} = 1$, one can write the Drude weight as

$$
D_{i,j} = \langle j_i, N^k\rangle\langle N_k, j_j\rangle,
\tag{S28}
$$

where $N_i = R_i^{\ j} Q_j$ is the total charge in the normal mode basis. Note that in the computation involving the current expansion (6), we always take the homogeneous limit of the averages only at the end of computations.

## S2.2 Derivation of the Onsager matrix

In this section, we present a derivation of the convective contribution to the Onsager matrix $\mathfrak{L}^c_{u,v}$, and hence the diffusion constant. Recall that the Onsager matrix is given by the following expression

$$
\mathfrak{L}_{u,v} = \int dt \left( \sum_x \langle j_u(x,t) j_v(0,0)\rangle^c - D_{u,v} \right),
\tag{S29}
$$

where $D_{u,v}$ corresponds to the Drude weight. In order to better understand how each term scale with $\ell$, let us rewrite it in terms of the hydrodynamic current $\mathsf{j}$

$$
\mathfrak{L}_{u,v} = \int d\tau \left( \sum_\chi \langle \mathsf{j}_u(\chi,\tau), \mathsf{j}_v(0,0)\rangle - D_{u,v} \right).
\tag{S30}
$$

We will take the hydrodynamic limit $\ell\to\infty$ only in the end. In order to study corrections to the Euler scale hydrodynamics, which corresponds to the nonzero Drude weight, and can be interpreted as a consequence of the first term in the expression for the current (S25), it proves useful to consider a part of the current that characterizes the sub-Euler contribution

$$
\mathsf{j}_i^-(\chi,\tau) = \mathsf{j}_i(\chi,\tau) - (\partial_{\mathsf{q}_j(\chi)}\langle \mathsf{j}_i(\chi)\rangle)\mathsf{q}_j(\chi,\tau).
\tag{S31}
$$

The Onsager matrix now reads

$$
\mathfrak{L}_{u,v} = \int d\tau \sum_\chi \langle \mathsf{j}_u^-(\chi,\tau), \mathsf{j}_v^-(0,0)\rangle.
\tag{S32}
$$

Inserting the expression of the current into equation (S30) we obtain

$$
\begin{aligned}
\mathfrak{L}_{u,v} &= \sum_\chi \frac{\ell}{2} \int d\tau (\partial_{\mathsf{q}_j(0,0)}\partial_{\mathsf{q}_i(0,0)}\langle \mathsf{j}_v(0,0)\rangle)(\langle \mathsf{q}_i(0,0)\mathsf{q}_j(0,0)\mathsf{j}_u(\chi,\tau)\rangle_n^c - \\
&\quad - A_u^k \langle \mathsf{q}_i(0,0)\mathsf{q}_j(0,0)\mathsf{q}_k(x,t)\rangle_n^c) = \\
&= \frac{\ell}{2}(\partial_{\mathsf{q}_j(0,0)}\partial_{\mathsf{q}_i(0,0)}\langle \mathsf{j}_v(0,0)\rangle)(M_{ij}^{\mathsf{j}_u} - A_u^k M_{ij}^{\mathsf{q}_k}),
\end{aligned}
\tag{S33}
$$

where we defined

$$M_{ij}^{\mathfrak{o}} = \sum_{\chi} \int d\tau \left\langle \mathsf{q}_i(0,\tau)\mathsf{q}_j(0,\tau)\mathfrak{o}(\chi,0)\right\rangle_n^c \tag{S34}$$

for an arbitrary hydrodynamic operator $\mathfrak{o}$, and introduced a normalized connected correlation function $\left\langle \mathsf{q}_i(0,0)\mathsf{q}_j(0,0)\mathsf{j}_u(\chi,\tau)\right\rangle_n^c \equiv \frac{1}{\ell}\left\langle \mathsf{q}_i(0,0)\mathsf{q}_j(0,0)\mathsf{j}_u(\chi,\tau)\right\rangle^c$. To proceed, let us first deal with a building block $M_{ij}^{\mathfrak{o}}$ and rewrite it in terms of normal modes. Using the solution of $\mathfrak{n}(\chi,\tau)$ (S24), we have

$$M_{ij}^{\mathfrak{o}} = (R^{-1})_i^{i'}(R^{-1})_j^{j'} \times$$
$$\times \sum_{\chi}\int d\tau \int_{-\pi}^{\pi}\frac{dk}{2\pi}\frac{dk'}{2\pi}e^{-i(k+k')\chi}e^{-i(\omega_{i'}(k))+\omega_{j'}(k'))\tau}\left\langle \mathfrak{n}_{i'}(0,0)\mathfrak{n}_{j'}(0,0)\mathfrak{o}(0,0)\right\rangle_n^c. \tag{S35}$$

Now using that $\sum_{\chi=-\infty}^{\infty}e^{ik\chi} = 2\pi\delta(k)$, we get

$$M_{ij}^{\mathfrak{o}} = (R^{-1})_i^{i'}(R^{-1})_j^{j'}\int d\tau \int_{-\pi}^{\pi}\frac{dk}{2\pi}e^{-i(\omega_{i'}(k)+\omega_{j'}(-k))\tau}\left\langle \mathfrak{n}_{i'}(0,0)\mathfrak{n}_{j'}(0,0)\mathfrak{o}(0,0)\right\rangle_n^c. \tag{S36}$$

The integration over $\tau$ can be done as follows

$$\int d\tau\, e^{-i(\omega_{i'}(k)+\omega_{j'}(-k))\tau} = \frac{2\pi\delta(k)}{|v_{i'}-v_{j'}|}, \tag{S37}$$

which allows us to obtain a compact expression of $M_{ij}^{\mathfrak{o}}$

$$M_{ij}^{\mathfrak{o}} = (R^{-1})_i^{i'}(R^{-1})_j^{j'}\frac{1}{|v_{i'}-v_{j'}|}\left\langle \mathfrak{n}_{i'}(0,0)\mathfrak{n}_{j'}(0,0)\mathfrak{o}(0,0)\right\rangle_n^c =$$
$$= (R^{-1})_i^{i'}(R^{-1})_j^{j'}\frac{1}{|v_{i'}-v_{j'}|}R_{i'}^{i''}R_{j'}^{j''}\left\langle \mathsf{q}_{i''}(0,0)\mathsf{q}_{j''}(0,0)\mathfrak{o}(0,0)\right\rangle_n^c =$$
$$= (R^{-1})_i^{i'}(R^{-1})_j^{j'}\frac{1}{|v_{i'}-v_{j'}|}R_{i'}^{i''}R_{j'}^{j''}\frac{\partial^2\langle o\rangle}{\partial\beta^{i''}\partial\beta^{j''}}. \tag{S38}$$

Notice that in the final line, the hydrodynamic observable $\mathfrak{o}(0,0)$ is replaced by the ordinary local observable $o$. We further observe that the curvature term $\frac{\partial^2\langle o\rangle}{\partial\beta^{i''}\partial\beta^{j''}}$ can be written as

$$\frac{\partial^2\langle o\rangle}{\partial\beta^{i''}\partial\beta^{j''}} = \frac{\partial^2\langle o\rangle}{\partial\mathsf{q}_k\partial\mathsf{q}_{k'}}C_{i''k}C_{j''k'} + \frac{\partial\langle o\rangle}{\partial\mathsf{q}_k}\frac{\partial}{\partial\beta^k}C_{i''j''}, \tag{S39}$$

according to which (S33) becomes

$$\mathfrak{L}_{u,v} = \frac{\ell}{2}(\partial_{\mathsf{q}_j(0,0)}\partial_{\mathsf{q}_i(0,0)}\langle \mathsf{j}_v(0,0)\rangle)(M_{ij}^{\mathsf{j}_u} - A_u^k M_{ij}^{\mathsf{q}_k}) =$$
$$= \frac{\ell}{2}(\partial_{\mathsf{q}_j(0,0)}\partial_{\mathsf{q}_i(0,0)}\langle \mathsf{j}_v(0,0)\rangle)(R^{-1})_i^{i'}(R^{-1})_j^{j'}\frac{1}{|v_{i'}-v_{j'}|}(R^{-T})_{i'k}(R^{-T})_{j'k'}H_u^{kk'} =$$
$$= \ell(\partial_{\mathsf{q}_j(0,0)}\partial_{\mathsf{q}_i(0,0)}\langle \mathsf{j}_v(0,0)\rangle)(R^{-1})_i^{i'}(R^{-1})_j^{j'}\frac{1}{|v_{i'}-v_{j'}|}(R^{-1})_u^{u'}G_{u'}^{i'j'}, \tag{S40}$$

where $H$ matrix corresponds to $H_v^{ij} = \ell\partial_{\mathsf{q}_j(0,0)}\partial_{\mathsf{q}_i(0,0)}\langle \mathsf{j}_v(0,0)\rangle$. The $G$-matrix is defined as

$$G_i^{jk} = \frac{1}{2}R_i^l\left(R^{-T}H_l R^{-1}\right)^{jk}. \tag{S41}$$

We are finally in the position to derive the exact convective contribution to the Onsager matrix. Putting everything together, we have

$$\mathfrak{L}_{u,v}^c = 2(R^{-1}\tilde{G}^2 R^{-\mathrm{T}})_{uv}, \tag{S42}$$

where

$$\tilde{G}_{ij}^2 = \frac{1}{|v_{i'} - v_{j'}|} G_{ii'j'} G_j^{i'j'}. \tag{S43}$$

Note that the convective Onsager matrix can also be written as

$$\mathfrak{L}_{u,v}^c = \langle j_u^- Q_i Q_j \rangle^c \mathcal{C}^{ij;i'j'} \langle Q_{i'} Q_{j'} j_v^- \rangle^c = \frac{\langle j_u^- N_i N_j \rangle^c \langle N^i N^j j_v^- \rangle^c}{2|v_i - v_j|}, \tag{S44}$$

where

$$\mathcal{C}^{ij;i'j'} = \frac{1}{2|v_{i''} - v_{j''}|} R^{i''i} R^{j''j} R_{i''}^{i'} R_{j''}^{j'}. \tag{S45}$$

## S2.3 Derivation of the $G$-tensor and diffusion in integrable systems

In integrable systems, quantum and classical ones, (Lieb-Liniger model, integrable spin chains, classical and quantum integrable field theories) as well as in gases of hard rods, thermodynamics can be systematically studied by thermodynamic Bethe ansatz (TBA). These models share the property that their dynamics is completely fixed by the 2-body scattering shift $T_{ij}$, which also provides the dressing for the thermodynamic functions. Dressing denotes the properties of the modes which are immersed in the background with the finite density of quasiparticles. In the thermodynamic limit any stationary state, thermal or GGE, is fixed by the occupation function $n_i = \langle \mathfrak{n}_i \rangle \sqrt{\chi_i}/\rho_i^{\mathrm{tot}}$ where the total density of states is given in terms of the occupations via an integral equation $\rho_i^{\mathrm{tot}} = p' + T n \rho_i^{\mathrm{tot}}$. Here $p_i'$ is the bare momentum of each quasiparticle $i$. The susceptibility of each mode is given by $\chi_i = \rho_i^{\mathrm{tot}} n_i (1 - n_i)$. Moreover the group velocities which we denote by $v_i^{\mathrm{eff}} = (\partial \varepsilon / \partial p)_i$ are again obtained by solving integral equation for the dressing of the energy $\varepsilon$ and momentum $p$. The label $i$ runs over the infinite number of distinct normal modes. In standard notations the index $i$ is labeled by the continuous parameter corresponding to rapidities $\theta$ and the discrete parameter labeling distinct quasiparticle types $s$.

An expression for the matrix $\mathfrak{L}$ was only recently found in using techniques of integrability in [22]. It takes the following form

$$\mathfrak{L}_{kl} = \left( R^{-1} \frac{\rho^{\mathrm{tot}}}{\sqrt{\chi}} \tilde{\mathfrak{D}} \frac{\sqrt{\chi}}{\rho^{\mathrm{tot}}} R^{-\mathrm{T}} \right)_{kl}, \tag{S46}$$

where $T_{ij} = T_{ji}$ is the scattering shift between modes, and $R_i^j = (1 - nT)_i^j / \sqrt{\chi_i}$. $n_i$ and $\rho_i^{\mathrm{tot}}$ are proportional to identity matrices. The diffusion kernel can be decomposed into diagonal and off-diagonal terms

$$\tilde{\mathfrak{D}}_{kl} = \delta_{kl} \sum_{k'} \chi_{k'} \left( \frac{T_{kk'}^{\mathrm{dr}}}{\rho_k^{\mathrm{tot}}} \right)^2 |v_k - v_{k'}| - \chi_k \frac{T_{kl}^{\mathrm{dr}} T_{lk}^{\mathrm{dr}}}{\rho_k^{\mathrm{tot}}} |v_k - v_l|, \tag{S47}$$

where $T_{ij}^{\mathrm{dr}}$ is the dressed scattering shift, given by the integral equation $T_{ij}^{\mathrm{dr}} - T_{ik} n_k T_{kj}^{\mathrm{dr}} = T_{ij}$. The dressed scattering phase shift $T^{\mathrm{dr}} = (1 - Tn)^{-1} T$ can be thought of as a length of the jump of the quasi-particle upon scattering with another quasi-particle, if both of them are immersed in a thermal background [23, 50]. This expression provides the diffusion constants of generic integrable chain, comprising systems of classical hard rods and spin chains [22, 31].

In order to derive this result using our expression

$$\mathcal{L}_{kl}^c = 2(R^{-1}\tilde{G}^2 R^{-\mathrm{T}})_{kl},\tag{S48}$$

we need to derive an explicit form of $G$-matrix in integrable systems. We will show that it reads

$$G_i^{kl} = \delta_i{}^k \mathfrak{g}_i{}^l + \delta_i{}^l \mathfrak{g}_i{}^k,\tag{S49}$$

where

$$\mathfrak{g}_i{}^k = \frac{T_{ik}^{\mathrm{dr}}\sqrt{\chi_k}(v_k - v_i)}{2\rho_i^{\mathrm{tot}}}.\tag{S50}$$

The convective coefficients in the normal mode basis is

$$\tilde{G}_{ik}^2 = \delta_{ik}\sum_{i'}\frac{2}{|v_i - v_{i'}|}\mathfrak{g}_i{}^{i'}\mathfrak{g}_{ii'} + \frac{2}{|v_i - v_k|}\mathfrak{g}_i{}^k \mathfrak{g}_{ki}.\tag{S51}$$

To see that (S51) when plugged into (S48) reproduces (S46), it is enough to check

$$\left(\frac{\rho^{\mathrm{tot}}}{\sqrt{\chi}}\tilde{\mathfrak{D}}\frac{\sqrt{\chi}}{\rho^{\mathrm{tot}}}\right)_{kl} = 2\tilde{G}_{kl}^2,\tag{S52}$$

which is obviously true, since the first term (diagonal) and the second term (off-diagonal) terms in (S51) precisely coincide with those in (S47) up to the factor 2. The factor 2 is then accounted for by the factor in front of $\tilde{G}_{kl}^2$ above.

Now we proceed to compute the $G$-tensor for integrable systems directly from known generalized hydrodynamics (GHD) expressions [43]. Note that the Latin indices $i = (\theta, a)$ denote the pairs of quasi-momentum $\theta$ and the particle type $a$, and the calculations boils down to simple matrix-like manipulations. To reiterate, the $G$-tensor reads

$$G_i = \frac{1}{2}R_i^l(R^{-1})^{\mathrm{T}}H_l R^{-1}, \quad \begin{cases} RAR^{-1} = \mathrm{diag}(v^{\mathrm{eff}}) \\ H_i^{jk} = \frac{\partial A_i^j}{\partial \rho_k} \\ RCR^{\mathrm{T}} = 1. \end{cases}\tag{S53}$$

We first normalize $R = \hat{\mathcal{N}}(1-nT)$ accordingly to the prescription $RCR^{\mathrm{T}} = 1$, using the normalization $\hat{\mathcal{N}}$. Since $C = (1-nT)^{-1}\rho(1-n)(1-Tn)^{-1}$, we see that appropriate normalization is provided by

$$\hat{\mathcal{N}}_i{}^j = \frac{\delta_i{}^j}{\sqrt{\rho_i(1-n_i)}} = \frac{\delta_i{}^j}{\sqrt{\chi_i}},\tag{S54}$$

where $\rho_i = \rho_i^{\mathrm{tot}}n_i$ and $\chi_i = \rho_i(1-n_i)$ is the quasi-particle susceptibility. The $G$-matrix becomes

$$G_i^{jk} = \frac{1}{2}\sqrt{\frac{\chi_j \chi_k}{\chi_i}}(1-nT)_i^{i'}((1-nT)^{-1})_{j'}^j H_{i'}^{j'k'}((1-nT)^{-1})_{k'}^k.\tag{S55}$$

Taking into account that

$$\frac{\partial}{\partial \rho_j} = \frac{\partial n_i}{\partial \rho_j}\frac{\partial}{\partial n_i} = \frac{n_i}{\rho_i}(1-nT)_i^j\frac{\partial}{\partial n_i},\tag{S56}$$

we have

$$G_i^{jk} = \frac{1}{2}\frac{n_j}{\rho_j}\sqrt{\frac{\chi_j \chi_k}{\chi_i}}(1-nT)_i^{i'}\frac{\partial A_{i'}^{k'}}{\partial n_j}((1-nT)^{-1})_{k'}^k.\tag{S57}$$

It is useful to decompose $\frac{\partial}{\partial n_j} A_i^{\ k}$ as follows

$$\frac{\partial}{\partial n_j} A_i^{\ k} = W_i^{\ jk} + Z_i^{\ jk}, \tag{S58}$$

where

$$W_i^{\ jk} = \Big[ \frac{\partial}{\partial n_j}((1-nT)^{-1})_i^{\ i'} \Big] v_{i'}^{\text{eff}}(1-nT)_{i'}^{\ k} + ((1-nT)^{-1})_i^{\ i'} v_{i'}^{\text{eff}} \frac{\partial}{\partial n_j}(1-nT)_{i'}^{\ k} =$$
$$= ((1-nT)^{-1})_i^{\ j} T^{jj'}(A_{j'}^{\ k} - v_j^{\text{eff}} \delta_{j'}^{\ k}), \tag{S59}$$

and

$$Z_i^{\ jk} = ((1-nT)^{-1})_i^{\ i'} \frac{\partial v_{i'}^{\text{eff}}}{\partial n_j}(1-nT)_{i'}^{\ k}. \tag{S60}$$

We first deal with the contribution coming from $W_{i'}^{\ jk'}$. Applying $(1-nT)_i^{\ i'}$ to it and summing over $i'$ results in

$$(1-nT)_i^{\ i'} W_{i'}^{\ jk'} = T_j^{\ j'}(A_{j'}^{\ k} - v_j^{\text{eff}} \delta_{j'}^{\ k}) \delta_i^{\ j}. \tag{S61}$$

Noting further that $A_k^{\ l}((1-nT)^{-1})_l^{\ j} = v_j^{\text{eff}}((1-nT)^{-1})_k^{\ j}$, we get

$$\frac{1}{2} \frac{n_j}{\rho_j}(1-nT)_i^{\ i'} W_{i'}^{\ jk'}((1-nT)^{-1})_{k'}^{\ k} = T_i^{\ i'} \frac{n_i}{2\rho_i}(v_k^{\text{eff}} - v_i^{\text{eff}})((1-nT)^{-1})_{i'}^{\ k} \delta_i^{\ j} =$$
$$= \frac{1}{2\rho_i^{\text{tot}}}(v_k^{\text{eff}} - v_i^{\text{eff}})(T^{\text{dr}})_i^{\ k} \delta_i^{\ j}. \tag{S62}$$

Let us next turn to the term involving $Z_i^{\ jk}$. We first recall that

$$\frac{\partial v_i^{\text{eff}}}{\partial n_j} = \frac{1}{n_j} \frac{\rho_j}{\rho_i}(v_j^{\text{eff}} - v_i^{\text{eff}})((1-nT)^{-1})_i^{\ j}, \tag{S63}$$

which leads to

$$\frac{1}{2} \frac{n_j}{\rho_j}(1-nT)_i^{\ i'} Z_{i'}^{\ jk'}((1-nT)^{-1})_{k'}^{\ k} = \frac{1}{2\rho_i}(v_j^{\text{eff}} - v_i^{\text{eff}})((1-nT)^{-1})_i^{\ j} \delta_i^{\ k} =$$
$$= \frac{1}{2\rho_i^{\text{tot}}}(v_j^{\text{eff}} - v_i^{\text{eff}})(T^{\text{dr}})_i^{\ j} \delta_i^{\ k}. \tag{S64}$$

Combining the results we finally end up with

$$G_i^{\ jk} = \frac{1}{2\rho_i^{\text{tot}}} \sqrt{\frac{\chi_j \chi_k}{\chi_i}} \Big[ (v_j^{\text{eff}} - v_i^{\text{eff}})(T^{\text{dr}})_i^{\ j} \delta_i^{\ k} + (v_k^{\text{eff}} - v_i^{\text{eff}})(T^{\text{dr}})_i^{\ k} \delta_i^{\ j} \Big] =$$
$$= \frac{1}{2\rho_i^{\text{tot}}} \Big[ \sqrt{\chi_j}(v_j^{\text{eff}} - v_i^{\text{eff}})(T^{\text{dr}})_i^{\ j} \delta_i^{\ k} + \sqrt{\chi_k}(v_k^{\text{eff}} - v_i^{\text{eff}})(T^{\text{dr}})_i^{\ k} \delta_i^{\ j} \Big] =$$
$$= \mathfrak{g}_i^{\ j} \delta_i^{\ k} + \mathfrak{g}_i^{\ k} \delta_i^{\ j}, \tag{S65}$$

where $\mathfrak{g}_i^{\ j}$ is given by (S50). Observe that (S65) is manifestly symmetric with respect to indices $j$ and $k$.

## S3 Quadratic lower bound in the normal mode basis

Here we relate the lower bound in terms of quadratic charges derived in [32] to the convective Onsager matrix $\mathfrak{L}_{kk}^c$. For simplicity we restrict the discussion to the infinite temperature state, however the generalization to finite temperatures should be possible by invoking exponential clustering property.

The first step in the derivation is to generalize the lower bound to multiple charges, by considering the norm of the operator $O = \frac{1}{T}\int_0^T \mathrm{d}t\, J(t) - \frac{\alpha_{i\geq j}}{L}N_i N_j$, on the finite lattice of length $L$

$$\langle A, B^\dagger \rangle = \frac{\mathrm{tr}(AB^\dagger)}{\mathrm{tr}(\mathbb{1})} - \frac{\mathrm{tr}(A)}{\mathrm{tr}(\mathbb{1})}\frac{\mathrm{tr}(B^\dagger)}{\mathrm{tr}(\mathbb{1})}, \tag{S66}$$

where we take the normal modes $N_i$ and the extensive current operator $J = \sum_x j_x$ to be traceless and Hermitian. The fact that $\langle O, O \rangle \geq 0$ gives us the lower bound

$$\frac{1}{T^2}\int_0^T \mathrm{d}t\mathrm{d}t' \langle J(t), J(t')\rangle \geq \frac{\alpha^{i\geq j}}{L}\langle J, N_i N_j\rangle + \frac{\alpha^{i\geq j*}}{L}\langle J, N_i N_j\rangle - \frac{\alpha^{i\geq j}\alpha^{k\geq l*}}{L^2}\langle N_i N_j, N_l N_k\rangle. \tag{S67}$$

Note that the three point function in the above expression is indeed equivalent to the three point connected correlation function due to the tracelesness of the operators, however the connected correlation involving four copies of normal modes corresponds to the two point connected correlation function of the terms in the brackets. In order to maximize the contribution on the right hand side we take a derivative with respect to $\alpha_{ij}^*$, which produces the set of equations

$$\frac{1}{L}\langle J, N_i N_j\rangle = \frac{1}{L^2}\alpha^{kl}\langle N_i N_j, N_l N_k\rangle. \tag{S68}$$

Taking into account the normal modes property

$$\frac{\mathrm{tr}(N_i N_j)}{\mathrm{tr}\,\mathbb{1}} = L \times \delta_{ij}, \tag{S69}$$

and the reduction (S16), we obtain the relation

$$\langle N_i N_j, N_l N_k\rangle = L^2(\delta_{ik}\delta_{jl} + \delta_{ij}\delta_{kl}) + \mathcal{O}(L). \tag{S70}$$

Inserting the property (S70) into the set of optimization conditions (S68) we obtain the set of coefficients

$$\alpha_{i,j} = \langle J, N_i N_j\rangle_n (1 - \tfrac{1}{2}\delta_{ij}). \tag{S71}$$

This yields a lower bound

$$\frac{1}{T^2}\int_0^T \mathrm{d}t\mathrm{d}t' \langle J(t), J(t')\rangle \geq \sum_{i\geq j}\langle J, N_i N_j\rangle^2 (1 - \tfrac{1}{2}\delta_{ij}). \tag{S72}$$

Finally the l.h.s of the above expression can be identified with the diffusion constant corresponding to the current $J$ [32]

$$\frac{1}{T^2}\int_0^T \mathrm{d}t\mathrm{d}t' \langle J(t), J(t')\rangle \propto D. \tag{S73}$$

Following [32], the lower bound reads

$$D \geq \frac{1}{8v_{LR}}\left(\sum_{i\geq j}\langle J, N_i N_j\rangle^2 (1 - \tfrac{1}{2}\delta_{ij})\right). \tag{S74}$$

Note that if we take into account the upper bound on the difference of two velocities $|v_i - v_j| \leq 2v_{LR}$ our prediction (S44) overshoots this lower bound by the factor of 4.

## S4 Remaining terms in operator expansion

There are two contributions in the operatorial expansion which scale as $\frac{1}{\ell}$ that we did not account for. The first one corresponds to conserved quantities $\delta\mathfrak{q}^{(2)}$ that scale as $\ell^2$, and which are not simply a product of local densities or a linear combination of the products, which by assumption means that $\langle\delta\mathfrak{q}^{(2)}, \delta\mathfrak{q}_i\delta\mathfrak{q}_j\rangle = 0$, implying that this contribution can be treated independently. Similarly, we can get $\frac{1}{\ell}$ scaling by choosing the first order in expansion, where one of the conserved charge densities lies in the neighboring cell of the operator which we are expanding. Once again such a contribution does not couple to the squares of local conserved densities, and can be treated on the separate footing.

Finally we should discuss higher order contributions from convective modes. First of all we conjecture that the correct expansion of local observables in terms of convective modes takes the following form

$$\delta\mathfrak{j}_k(\chi,\tau) = (\partial_{\mathfrak{q}_i(\chi)}\langle\delta\mathfrak{j}_k(\chi,\tau)\rangle)\delta\mathfrak{q}^i(\chi) + \frac{1}{2}(\partial_{\mathfrak{q}_j(\chi)}\partial_{\mathfrak{q}_i(\chi)}\langle\delta\mathfrak{j}_k(\chi,\tau)\rangle)\delta(\mathfrak{q}^i(\chi)\mathfrak{q}^j(\chi)) +$$
$$+ \frac{1}{6}(\partial_{\mathfrak{q}_k(\chi)}\partial_{\mathfrak{q}_j(\chi)}\partial_{\mathfrak{q}_i(\chi)}\langle\delta\mathfrak{j}_k(\chi,\tau)\rangle)\delta(\mathfrak{q}^i(\chi)\mathfrak{q}^j(\chi)\mathfrak{q}^k(\chi)) + ...,$$

where the expansion satisfies the natural property

$$\langle\mathfrak{o}, \delta(\mathfrak{q}^i(\chi)\mathfrak{q}^j(\chi)\mathfrak{q}^k(\chi))\rangle = \langle\mathfrak{o}\mathfrak{q}^i(\chi)\mathfrak{q}^j(\chi)\mathfrak{q}^k(\chi)\rangle^c. \tag{S75}$$

This form produces a correct scaling of any multipoint connected correlation function.

Following our conjecture we now elaborate on possible higher order contributions to the Onsager matrix from the hydrodynamical expansion of the current. For simplicity we will consider only the third order correction. Following the same steps as in the derivation of the Onsager matrix from the second order, we obtain the contribution

$$\mathcal{L}_{v,u} = \frac{\ell^2}{6}(\partial_{\mathfrak{q}_r(0,0)}\partial_{\mathfrak{q}_j(0,0)}\partial_{\mathfrak{q}_i(0,0)}\langle\mathfrak{j}_v(0,0)\rangle)(M_{ijr}^{\mathfrak{j}_u} - A_u^k M_{ijr}^{\mathfrak{q}_k}) + \mathcal{O}(\ell^{-1}), \tag{S76}$$

with

$$M_{ijr}^{\mathfrak{o}} = \frac{1}{\ell}\sum_{\chi}\int d\tau \left\langle \mathfrak{q}_r(0,\tau)\mathfrak{q}_i(0,\tau)\mathfrak{q}_j(0,\tau)\mathfrak{o}(\chi,0)\right\rangle_n^c. \tag{S77}$$

Going to the normal mode basis, we obtain

$$M_{ijr}^{\mathfrak{o}} = \frac{1}{\ell}(R^{-1})_r^{\ r'}(R^{-1})_i^{\ i'}(R^{-1})_j^{\ j'} \times$$
$$\times \sum_{\chi}\int d\tau \int_{-\pi}^{\pi}\frac{dk}{2\pi}\frac{dk'}{2\pi}\frac{dk''}{2\pi}e^{-i(k+k'+k'')\chi}e^{-i(\omega_{i'}(k)+\omega_{j'}(k')+\omega_{r'}(k''))\tau} \times$$
$$\times \left\langle \mathfrak{n}_{i'}(0,0)\mathfrak{n}_{j'}(0,0)\mathfrak{n}_{r'}(0,0)\mathfrak{o}(0,0)\right\rangle_n^c. \tag{S78}$$

Summation over $\chi$ and integration over time produces

$$M_{ijr}^{\mathfrak{o}} = \frac{1}{\ell}(R^{-1})_r^{\ r'}(R^{-1})_i^{\ i'}(R^{-1})_j^{\ j'} \times$$
$$\times \int_{-\pi}^{\pi}\frac{dk}{2\pi}\frac{dk'}{2\pi}\frac{dk''}{2\pi}2\pi\delta(k+k'+k'')2\pi\delta((\omega_{i'}(k)+\omega_{j'}(k')+\omega_{r'}(k''))) \times$$
$$\times \left\langle \mathfrak{n}_{i'}(0,0)\mathfrak{n}_{j'}(0,0)\mathfrak{n}_{r'}(0,0)\mathfrak{o}(0,0)\right\rangle_n^c. \tag{S79}$$

Let's assume that $v_i \geq v_j \geq v_r$. Integrating over $k''$, we get

$$
M^{\mathfrak{o}}_{ijr} = \frac{1}{\ell}(R^{-1})^{r'}_r (R^{-1})^{i'}_i (R^{-1})^{j'}_j \times
$$
$$
\times \int_{-\pi}^{\pi} \frac{\mathrm{d}k\,\mathrm{d}k'}{2\pi} \delta((\omega_{i'}(k) + \omega_{j'}(k') + \omega_{r'}(-k'-k))) \times
$$
$$
\times \left\langle \mathfrak{n}_{i'}(0,0)\mathfrak{n}_{j'}(0,0)\mathfrak{n}_{r'}(0,0)\mathfrak{o}(0,0) \right\rangle^c_n . \tag{S80}
$$

Integration over $k$ and $k'$ finally yields

$$
M^{\mathfrak{o}}_{ijr} = \frac{1}{\ell}(R^{-1})^{r'}_r (R^{-1})^{i'}_i (R^{-1})^{j'}_j \frac{1}{v_i - v_r} \left\langle \mathfrak{n}_{i'}(0,0)\mathfrak{n}_{j'}(0,0)\mathfrak{n}_{r'}(0,0)\mathfrak{o}(0,0) \right\rangle^c_n . \tag{S81}
$$

Obviously the contribution vanish in the limit $\ell \to \infty$, provided that the degeneracies are absent.

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
