# Peer review of "Diffusion from Convection"

_SciPost Physics, doi:SciPost Phys. 9, 075 (2020)_

## Round 3 · Referee Report · Christian Mendl (Referee 1) · 2020-9-2

Strengths

  1. Clear and detailed mathematical exposition of the overall framework
  2. Derivation of the "magic formula"
  3. Well written, and well chosen notation

Report

The work provides valuable insights and an overarching framework to understand transport properties of physical systems. Specifically, the authors use a second order expansion of the current to derive the contribution to diffusion by ballistic modes in Eqs. (11) and (12) via the Onsager matrix. En passant, the work is able to re-derive and explain some previous results in the literature, like the mentioned "magic formula", and relate the results to previous work on integrable models.

---

## Round 3 · Referee Report · Anonymous (Referee 2) · 2020-10-5

Strengths

  • Interesting and timely subject
  • Clearly written and results clearly stated

Weaknesses

  • Too synthetic in certain parts of the main discussion, in particular the section on the “magic Formula”.

Report

This is a good paper, in my opinion, presenting a quite general approach to discuss hydrodynamic transport in generic many-body systems. The approach taken is to discuss the general ideas, their physical origin and their implications (lower bounds on diffusion constant. and magic formula) in the main text relegating all technical details in the appendices. I find the results quite interesting and the style of presentation overall satisfactory for a subject that could easily become heavily technical. In my opinion the broad community interested in hydrodynamics of many-body systems will find this paper quite useful and it therefore meets the criteria for publication in SciPost Physics.

---

## Editorial Decision

published